# Will China's audit of natural environmental resource promote green sustainable development? Evidence from PSM-DID analysis based on substantial and strategic pollution reduction

**Guofeng Li**[1☯], **Meng Wu**[2]*, **Ruoyuan Sun**[3☯]

1 School of Statistics and Mathematics, Shandong University of Finance and Economics, Jinan, China,
2 School of Economics, Shandong University of Finance and Economics, Jinan, China, 3 School of Public Finance & Taxation, Shandong University of Finance and Economics, Jinan, China

☯ These authors contributed equally to this work.
* wumeng0528@126.com

**Data Availability Statement:** All relevant data are within the paper and its Supporting Information files.

## Abstract

Countries around the world have become concerned about their responsibility to protect the environment and resources. In this paper, we established a model of multi-period PSW-DID (weighted propensity score matching- differences-in-differences) to test the effect of China's audit of natural resource. We found that: (1) local governments had a strategic incentive to reduce pollution, they paid more attention to environmental performance linked to individual promotion than to green innovation and development. (2) Compared with the long-term and complexity of water pollution control, they paid more attention to air pollution treatment. (3) In the long run, the environmental audit was indeed beneficial to the improvement of environmental quality, but the time of taking effect was the second year (one-year lag). (4) In addition, because of the contradiction between the neglect of human capital and the timeliness of environmental supervision, the local government did not show substantial pollution reduction. Therefore, local officials should foster the long-term responsibility consciousness of green innovation and pay more attention to the integration of human capital. The audit of natural resources should establish a long-term mechanism, which could establish a complete accountability system or change off-office audit to interim audit. The construction of audit big data platforms should pay more attention to substantive characteristic data, such as data on population inflow, which is not only a paper score of air pollution. This study can reveal the dilemma of pollution prevention and control in China, urge local governments to promote the rational flow of human resources, improve the innovation level, and achieve substantive pollution control and efficiency enhancement of green development.

## 1. Introduction

In September 2019, wildfires broke out in Australia's forests; In December 2019, COVID-19 spread; In January 2020, Mt. Everest was covered with vegetation; In February 2020, the

**Funding:** The author(s) received no specific funding for this work.

Antarctic ice melted. Such extended risks may seem unrelated, but the common connotation is a wake-up call for the world to deal with environmental problems. The Chinese government has always advocated a green approach. The report of the 19th National Congress of the Communist Party of China listed pollution prevention and control as one of the three critical battles to complete the building of a moderately prosperous society in all respects and called for efforts to address prominent environmental problems. General Secretary Xi also duly put forward the judgment that "clear waters and green mountains are mountains of gold and silver". In addition, the two sessions in 2020 once again emphasized that we should continue to promote pollution prevention and control and maintain the determination of green development.

China is also implementing green development and gradually adjusting its economic development model, from extensive growth with high energy consumption and high pollution to efficient growth with innovation as the core [1]. Green development requires both economic growth and environmental protection. In terms of its effect, not only efforts should be made on "pollution reduction", but also sustainable "efficiency enhancement" should be realized. In general, pollution reduction is mainly considered from two dimensions: source and terminal. Source pollution reduction is the structure and scale of control factors input, especially high-pollution factors input; Terminal decontamination is to treat the generated pollutants and reduce emissions [2]. However, in terms of the combination of substantial efficiency enhancement and pollution reduction, more attention should be paid to process improvement. That is, scientific and technological innovation in the production process should be realized, independent research and development should be promoted, pollution emissions should be reduced, and production efficiency should be improved [3].

The core of innovation-driven green development lies in perfecting environmental regulations. The experience of foreign developed countries, mostly focuses on the tax system, covering pollution tax, resource tax, energy tax, and traffic tax [4]. China is also gradually adjusting its environmental governance policies, for example, the Implementation of the Environmental Protection Tax Law in 2018. For a long time, problems such as insufficient rigidity of law enforcement, more administrative intervention, and lack of compulsion and standardization have formed an internal constraint mechanism for reducing pollution and increasing efficiency. The GDP-oriented tournament promotion assessment mode of local officials has also triggered path dependence for economic growth at the expense of resources and the environment [5]. In response to this phenomenon, the Chinese government proposed to "exploration should be made of the balance sheet of natural resources, audit leading officials on natural resources assets when they leave office and establish a lifelong accountability system for ecological environmental damage". It is committed to using the "reverse force" mechanism to encourage leading officials to establish the ecological civilization concept of respecting, complying with and protecting nature. Since 2015, audits of natural resource assets had been carried out nationwide, mainly involving key areas such as air pollution, land resources, water resources, forest resources, and mine ecological environment [6].

Academics have put forward many important views on innovative development and environmental protection. Some researchers believe that protecting the ecological environment is to protect productivity, and improving the ecological environment is to develop productivity [7]. Such research links productivity with environmental protection, and innovation is the most critical factor to improve productivity [8]. Another researcher, starting from the influencing factors between the two, believes that green technology innovation is conducive to the emission reduction effect of manufacturing industry in the high human capital group, but the emission reduction effect on the low human capital group is not significant, mainly because human capital and green technology innovation have a certain synergistic effect on reducing

environmental pollution [9]. As previously noted, most studies focus on the relationship between innovation and environmental protection or the influencing factors of both.

Although China's audit of natural resource is an environmental policy, its implementation is not only reflected in the effective regulation of environmental issues, but its operation is more likely to present a harmonious development approach of local governments for environmental protection and economic development. However, local governments in China are subject to environmental and economic performance assessment pressures and tend to show different motivations in dealing with pollution problems. They show strategic pollution reduction motivations rather than substantive pollution control and efficiency, and leadership cadres focus more on promotion-linked environmental performance than on innovative development performance. Most researchers have explored the issue from the perspective of pollution control and classification of pollution factors [10], and few have analyzed the behavior of local governments in response to environmental audits from a motivational perspective. In fact, in addition to responding to the pollution control audit, local governments also engage in environmental protection activities to obtain other benefits, which is a strategic behavior. Studying the motives of different environmental protection has important academic value and policy significance for understanding governmental behavior in the audit implementation process, evaluating the effect of policy implementation, and sorting out the mechanism of action. From the perspective of motivation, is it strategic or substantive behavior that local governments take when responding to the audit of natural resource? Are the policy effects of audit supervision realized in the current period or in the long term? We will devote ourselves to the discussion of this issue in this study.

According to the national policy, 2015 was the initial stage of the outgoing audit of natural resource assets, and a complete audit system was established by 2020. The audit system timely echoes the national concept of green development, which provides a rare "natural experiment" for this paper. Therefore, this paper takes the outgoing audit of natural resource assets in 2015 as a starting point and attempts to conduct a quasi-natural experimental evaluation of its green development effect. Specifically, the paper uses motivation as a clue to analyze the disposal policies of local governments for environmental problems, that is, the choice of station substantial efficiency or strategic pollution reduction, and on this basis, to explore the mechanism of this effect. This study is helpful to enrich the research content of audit of natural resources and green development in theory, and provide a reference for improving government-enterprise interaction and effectively evaluating government behavior in practice.

This paper may contribute in the following aspects: First, studying the effect of environmental audits in guiding local governments to promote environmental protection not only enriches the research on the effect of policy implementation, but also allows us to examine the long-term effect of audits based on the perspective of green development; Secondly, examining the effects of macro policy implementation from the perspective of government motives expands the study of economic policy and government behavior, not only providing new empirical evidence but also further examining the mechanism of its impact, which helps to understand the transmission mechanism between economic policy to local government behavior; Finally, an analysis based on the presence of government regulation and strong intervention can deepen the understanding of local environmental protection and governance effects. Moreover, the limitations of the study are that the data structure is not really able to support the pollution data of individual enterprises, and that large-scale clusters of the same type of production have not yet been formed in various regions of China, and that the pollution situation is not limited to the scope of the enterprises as the weather changes, so we have to choose the region—the prefecture-level city—as the starting point for sample selection.

## 2. Relevant theoretical analysis

The academic circles generally continue to explore government regulations and pollution prevention from four perspectives: policy attributes, development concepts, pollution characteristics, and supervision framework. The specific expansion is as follows:

### 2.1 Heterogeneity of environmental regulation: Substantive and strategic trade-offs

The essence of environmental regulation lies in how to deal with the negative externalities of pollution, which means regulating various social subjects with the help of relevant systems and policies to promote the realization of environmental protection goals. Rather than reducing pollution, the central government pays more attention to the coordination between environmental quality and economic growth, that is, the parallelism of green and development. It is worth noting that environmental performance assessment will be disregarded or discounted under the decentralized management system [11]. For political promotion, local governments compete in pursuit of GDP, which leads to the sacrifice of some resources and the environment [12].

From the perspective of social and environmental domains, some researchers have found that increased local financial pressure significantly reduces the efficiency of local governments' environmental governance, and also increases the conflict between different public decision values of the government, and increases the complexity of governance [13]. Relevant studies also concluded that there is still a significant positive relationship between public concerns and demands and regional environmental governance efficiency, and that the positive relationship between public concerns and demands and regional environmental governance efficiency is more significant in remote areas where the local government's direct intervention in governance is weaker. It also indicates that there is an elasticity in the efficiency of environmental governance when local governments respond to environmental problems [14].

Furthermore, this paper argues that due to limited financial resources and the accountability of cadre performance appraisal, local governments will be stuck between strategic and substantive environmental disposal choices. In other words, from the perspective of motivation, they will have to choose between pollution reduction and efficiency improvement.

### 2.2 Stages of green development: Factor-oriented and innovation-oriented

China's economic growth has shown stages. Since the reform and opening-up, China's comparative advantage has been characterized by high savings, high investment, high consumption, high emissions, labor-intensive and export-oriented, which is called "unsustainable growth" [15]. Since then, the Scientific Outlook on Development emphasizes sustainable development and requires the reform of the extensive growth model of factor input and the transformation to the intensive model of low consumption, low emissions, low pollution, and high efficiency [16]. The economic development in the new era reiterates the concept of green, in which innovation becomes the first driving force of green development [17], and is committed to the harmonious unity of development and protection. The phased nature of development does not mean the rejection of previous progress, but protection and improvement based on inheritance. The choice of the green path shows the difficult process of China's economy becoming innovation-driven.

Most researchers believe that, in the stage of economic and social development, for provinces with mature economic development and concentrated factors, the key to improve policies is to adopt preferential tax policies to encourage the development of high-tech enterprises,

build technology innovation funds for small and medium-sized science and technology enterprises, establish venture capital mechanisms and other measures to promote technological innovation and improve the technical level of factor resource utilization to achieve the transformation of the green development model [18]; While for provinces with lagging economic development with factor concentration, the focus of factor efficiency improvement policies is to reduce excessive government intervention in the allocation of factor resources [19]. This shows that whether the factors can be reasonably used is related to the green innovation model and the economic growth mode of an economy.

## 2.3 Particularity of pollution control: Short-term and long-term test

In the short term, in pursuit of higher profits, some enterprises will squeeze innovation input and pollute by extraction or distribution, thus prolonging the production technology research and development cycle and attenuating the innovation power [20]. However, According to Porter's hypothesis, investment in pollution control has a negative effect in the short term, but in the long run, it promotes technological progress significantly [21]. Moreover, when long-term passive pollution control has a high cost, enterprises will try to improve the treatment effect through technological innovation, which is the so-called innovation compensation mechanism [22]. Further, with the strengthening of government environmental regulations and the law of diminishing marginal performance of existing technologies, enterprises will increase the level of R&D investment and promote the acceleration of technological innovation [23].

Environmental pollution treatment requires long-term investment, but the trend of low and slow growth in treatment effectiveness makes it difficult to effectively improve the cost-efficiency ratio of treatment investment [24]. Thus, it causes double pressure on local governments for financial expenditure and performance improvement [25]. However, local governments have to consider the contradictory nature of party and government leaders' tenure and economic performance assessment in pollution governance, which leads to a hesitation between the short-term nature of leadership tenure and the long-term nature of pollution control. In addition, some studies have put forward opposite results. They believe that, as a typical mobile environmental governance policy, the central environmental protection supervision can have a long-term positive impact on pollution control, which may be due to the fact that the central environmental protection supervision has changed from one-time law enforcement to routine law enforcement. That is to say, there is a time conflict in the process of pollution control by local governments. Local governments will comprehensively consider short-term and long-term decisions from the perspective of the particularity of pollution control [26].

## 2.4 Comprehensive environmental audits: Integration of full coverage and period

The State Council has improved the outgoing audit of natural resources and assets, which has been taken seriously by governments at all levels as an innovative system for ecological civilization construction [27]. This regulation is devoted to supervising and inspecting the leading cadres' compliance with regulations and promoting the strengthening of environmental supervision responsibilities [28]. Audit of natural resource assets involves the environmental category, which is an important part of realizing the full coverage of audit supervision [29]. However, environmental governance is by no means accomplished overnight and requires long-term mechanisms to guarantee it [30]. Audit supervision has long assumed long-term responsibilities, such as the periodicity of economic responsibility audit, the annual nature of fiscal revenue and expenditure audit, and the tracking nature of poverty alleviation audit.

The biggest difficulty in environmental auditing is that the audit elements encompass too wide a range and span too long a time. For example, in terms of environmental quality, it covers air, water resources, etc., while air contains $CO_2$, $SO_2$, etc. [31]. And the long-term nature of environmental governance also means that environmental protection supervision must not only rely on the time point of the end of the current audit but should track the audit for a long time to achieve full coverage and integration of the whole period [32]. Therefore, this paper believes that the audit system, especially environmental audit, should realize the full coverage of the horizontal field and the whole period of vertical prescription integration.

## 2.5 Above literature analysis

A review of the above works of literature shows that although they are highly innovative in their respective fields, there is still room for improvement. First, although the heterogeneity of environmental regulations is fully demonstrated, the emphasis is on regulation effects, and the long-term effects of policies are not considered from the perspective of motivation. Second, although the stage of green development is studied in detail, it focuses on model transformation and lacks the incentive effect of policy from the perspective of utility. Third, although the particularity of pollution control is precisely discussed, it focuses on the performance of enterprises and lacks the initiative to test governance from the perspective of the government. Fourth, although it fully interprets the comprehensiveness of audit supervision, it focuses on horizontal coverage and lacks a long-term mechanism to examine policies from the perspective of time. In addition, different audit fields lead to different strategies of local governments, and there is a slight lack of research on the evaluation of the effects of specific laws within the audit system framework. Given the current research status, this paper aims to investigate the strategic choice of local governments to deal with audit of natural resource assets, and comprehensively understand the long-term mechanism of environmental audit from the perspective of motivation.

# 3. Model construction and variable description

## 3.1 Effect assessment model

Compared with water pollution and soil pollution, air pollution is not only the most intuitive and concerned, but also easier to observe, identify and measure [33]. As a result, local government will present motivation heterogeneity in dealing with air, water, and soil pollution. In addition, the connotation of green development is not only about reducing pollution but also about enhancing efficiency [34]. That means how to improve quality and efficiency within the framework of environmental protection. Therefore, by considering the effects of air quality improvement and water pollution control, this paper explores the specific effectiveness of audit of natural resource assets in the performance assessment of leading cadres' environmental responsibility and economic responsibility.

The extension of green development involves many factors and the timeliness of audit supervision conflicts with the short-term tenure of leading cadres (3 years on average). This shows that it is insufficient to evaluate the environmental and economic effects only based on the comparison of the mean values before and after the pilot audit. Furthermore, to eliminate exogenous factors such as natural endowment and ensure the sequence of audit pilot time points, to effectively carry out a quasi-natural experimental evaluation, this paper intends to adopt a multi-stage differential model (DID), that is, to identify the utility differences between the treatment group and the control group based on the endogenous conditions of regulation.

The double difference method is also called the "multiplicative difference method", which is abbreviated as DID, from the full English term "Differences-in-Differences", and has been

widely used in policy evaluation in recent years for the following reasons: (1)The endogeneity problem can be avoided to a large extent: policies are generally exogenous to microeconomic agents, and thus there is no reverse causality problem. Moreover, the use of fixed effects estimation alleviates the problem of omitted variable bias to a certain extent. (2) In contrast to the traditional method, which assesses policy effects by setting a dummy variable for the occurrence or non-occurrence of policies and then running a regression, the model setting of the double difference method is more scientific and can estimate the policy effects more accurately. Based on this, this paper intends to set the model (DID) as follows:

$$y_{it} = \beta_0 + \beta_1 d\mu_{it} + \beta_2 dt_{it} + \beta_3 du_{it} dt_{it} + \beta_4 Z_{it} + \delta_i + \sigma_t + \varepsilon_{it}$$

Among them, subscripts $i$ and $t$ indicate the region and year of the outgoing audit pilot of natural resource assets. $Z$ is the relevant control variable. The random perturbation term is $\varepsilon$ $du$ and $dt$ are used as two dummy variables, to distinguish the treatment group and control group before and after the pilot audit. $du = 1$ represents the pilot area, $du = 0$ is not piloted area. $dt = 1$ represents the year after the pilot started, $dt = 0$ is the year before the pilot. $\beta_3$ is the policy coefficient focused on by the dual difference method in this paper. The specific environmental and economic policy effects of outgoing audit of natural resource assets are reflected by this coefficient. $\delta_i$, $\sigma_t$ are used to control region fixed effect and year fixed effect respectively.

The principle of DID is to select two groups of similar samples, one is affected by the policy and the other is not. Identify the impact of policies by comparing the differences between the two before and after being affected by policies. The two groups of samples need not be very similar, as long as they have the same trend. DID deals with endogeneity. First, I have found two groups of individuals with the same trend. The only difference between the two groups is that one group is affected by the policy and the other is not. So this is close to a randomized trial.

To satisfy the premise of DID model——the parallel trend hypothesis, that is, to control the systematic difference and selectivity bias generated over time, this paper also constructed a PSW-DID model (weighted propensity score matching multiple differences method). On the one hand, PSW matches observable variables and controls common trends by weighted grouping to ensure exogenous policies. On the other hand, DID is used to overcome the limitations of PSW on unobservable variables. Among them, the PSW-DID part is mainly expanded according to the following parts: ① To estimate $\hat{y}$ in sequence using Logit regression and assign Each group the weight to $\hat{y}/(1 + \hat{y})$, and get the specific matching score value; ② To estimate the changes of environmental and economic effects before and after the pilot audit; ③ To obtain the average treatment effect (ATT) by subtracting the two groups of variation values.

## 3.2 Variable description and data source

**3.2.1 Data sources.**   In this paper, the air quality index data of prefecture-level cities were selected to match the urban invention and innovation data, the extreme values were eliminated, and the continuous variables were winsorized by 1% and 99%. Meanwhile, some missing data were manually queried and input. Among them, the pilot information about the outgoing audit of natural resources assets (pilot region and pilot time) was obtained from the website of the National Audit Office and the website of provincial and municipal audit institutions, and relevant information and environmental audit departments were consulted. The data of urban invention patents were selected from The Statistical Yearbook of High-tech Industry and the SIPO patent database of The State Intellectual Property Office of China. Meanwhile, the panel data of "annual city number of invention patents" was constructed by

referring to the practice of Heetal. The AQI is mainly derived from the National Urban Air Quality Daily, while some missing data and water pollution control data are collected and supplemented manually by environmental protection bureaus and meteorological bureaus. The control variables are mainly the specific variables affecting the environment, economy, and innovation as well as the characteristic variables of officials. They are selected from the data published in the Statistical Bulletin of National Economic and Social Development, Xinhuanet, People's Daily Online, and other authoritative websites, and partly obtained from the internal public information of the National Audit Office. Due to data integrity, Hong Kong, Macao, and Taiwan were excluded, and missing years in some cities were also excluded.

**3.2.2 Escription.** (1) Environmental indicators. Air Quality Index (Aqi): this paper takes AQI as the measurement index of air pollution and constructs the mean value, peak value, and improvement value of the urban AIR quality index for comparison. Air quality is divided into six grades: excellent, good, mildly polluted, moderately polluted, severely polluted, and severely polluted. The higher the AQI value is, the more serious the pollution is. In addition, air quality reflects the characteristic that spring and winter seasons are higher than summer and autumn seasons. Compared with the mean indicator of annual air quality in each city, the peak indicator not only reflects the pollution level but more importantly, compared with other statistical characteristic values, the peak indicator is more easily perceived by the public, so we choose the peak air quality (Mqi) as another reflection of air quality. The city-level water treatment capacity (Citysewage): through the summary of relevant sewage treatment test site data, to comprehensively measure the sewage treatment and improvement capacity of prefecture-level cities.

(2) Economic indicators. Economic performance and macroeconomic development level are the core of green development. This paper uses the logarithmic form of GDP per capita (Lnpgdp) and GDP growth rate (Rgdp) to describe this indicator. The number of invention patent applications at the city level (Citypatent) is used to describe the innovation capability of prefecture-level cities by summarizing individual invention patent data. The number of urban population and density of the urban population, as well as human resources, have always been considered the key decisive factor for the improvement of the economic level. Limited by the availability of population flow data at the city level, this paper adopts the number of the urban statistical population (Population) and density of urban population (Popdst) to conduct objective measurement.

(3) Other variables: To fully interpret the fitting effect of the composite group and ensure the robustness of the estimated results, some variables covering natural characteristic variables, economic characteristic variables, and official information were added: Concentration of six single pollutants ($SO_2$, $PM10$, $PM2.5$, $CO$, $NO_2$, $O_3$), and the following control variable saverage temperature (Temperature), precipitation (Rainfall), humidity (Humidity), sunshine duration (Sunshine), logarithmic form of consumer price index (Lncpi), age of mayor (Age), education background (Edu), promotion probability of officials (Promt), whether the first year in office (Tenure). Moreover, in the parameter estimation, we will control the time dummy variable and the region dummy variable. See S1 Appendix for relevant descriptive statistics.

# 4. Empirical results and robustness test

## 4.1 Benchmark evaluation of green development effect

In this paper, a multi-period dual differences model is constructed to evaluate the green development effect of outgoing audit of natural resources, as shown in Table 1 below. Among them, the differences between groups are set as follows: The explained variables in Equations (1) and (2) are the air quality index (Aqi) value of prefecture-level cities, (3) and (4) are listed as sewage

**Table 1. Baseline evaluation of green development effect.**

| The variable name | (1) Aqi | (2) Aqi | (3) Citysewage | (4) Citysewage | (5) Citypatent | (6) Citypatent |
|---|---|---|---|---|---|---|
| The current effect | -0.0566 (-1.3381) | -0.0571 (-1.1596) | 0.0099 (0.6430) | 0.0110 (0.8525) | -0.0083 (-0.4300) | -0.0025 (-0.1269) |
| Delayed stage effect | -0.1388* (-1.9770) | -0.1074* (-1.7816) | 0.0564 (1.0727) | 0.0379 (0.8375) | -0.0526 (-1.3093) | -0.0635 (-1.4570) |
| Control variables | | control | | control | | control |
| Time fixed effect | control | control | control | control | control | control |
| Urban fixed effect | control | control | control | control | control | control |

Note: ***, ** and * are significance level tests with significance of 1%, 5% and 10% respectively. The brackets are standard errors of the estimated values. Limited by the length of the table, please refer to the appendix and annex for detailed variable parameter estimates.

treatment capacity, and model (5) and (6) are the number of city patent applications; Intragroup differences were set as controls for whether or not a variable was introduced. Under normal circumstances, leading cadres will serve for at least two years (the first year is the probation period). The lag period is introduced in this paper to calibrate the lag effect of the policy experiment period and control the endogeneity.

According to the relevant estimates of the air quality index in Table 1, whether control variables are added or not, the current effect of the audit pilot is not significant, but the lag phase passes the significance test at the level of 10%, and the negative coefficient indicates the improvement of air governance. At the same time, the trend of lag effect indicates that for natural resource assets audit, the immediacy of leaving office should be abolished, a long-term mechanism should be established, and the accountability system can be established or the audit mode can be changed as the mid-term audit, to realize the responsible supervision of air quality of leading cadres during their specific tenure.

In addition, the relevant estimates of sewage treatment capacity were not significant, indicating that the audit pilot did not improve the local water pollution treatment capacity and the leading cadres did not effectively control water resources. Compared with air treatment, the water treatment process is longer. Meanwhile, the water pollution problem is more complex, which is difficult to be perceived by the public and less likely to produce informal environmental regulation pressure. The long-term nature of water pollution conflicts with the short tenure of leading officials. Local governments pay more attention to the treatment of pollutants with immediate results, such as air pollution, to achieve a link with their regulatory responsibilities during their term of office. On the one hand, pollutant attributes are more heterogeneous for natural resource asset audit and should not be treated equally. On the other hand, it also shows that leading cadres have subjective selectivity when considering economic responsibility and environmental responsibility, and there is a superficial strategic motivation to reduce pollution.

Green development requires the effective integration of economic performance and environmental performance, and high-quality economic development cannot be separated from the improvement of innovation ability. In this paper, the number of patent applications at the city level is used to test the innovation and development effect of environmental audit, and the estimated results are shown in Formula (5) and (6) in Table 1. Among them, the correlation coefficients of the current period and lag period did not pass the significance test, indicating that the audit of natural resource assets did not effectively improve innovation ability, and leading cadres did not attach substantial importance to economic development performance. In other words, despite advocating green development all the time, local governments show strategic pollution reduction rather than substantive pollution improvement due to environmental promotion criteria.

## 4.2 Robustness test

The parallel trend test, triple differences test (DDD), Placebo test (Placebo test), and weighted propensity matching score (PSW-DID) were used to ensure the robustness of the multi-stage DID model estimation.

**4.2.1 Trend test.** The principle of DID is to select two groups of similar samples, one is affected by the policy and the other is not. Identify the impact of policies by comparing the differences between the two before and after being affected by policies. The two groups of samples need not be very similar, as long as they have the same trend. This is the most important parallel trend assumption of DID. Two groups of individuals with the same trend are found. The only difference between the two groups is that one group is affected by the policy and the other is not. So this is close to a randomized trial. Therefore, we conduct a common trend test.

To avoid grouping affected by systematic differences between the economy and natural endowment of some cities before the pilot audit, a parallel trend test is used in this paper to verify whether the design requirements of the dual difference model are met. As shown in Table 2 below, before the audit pilot in 2015, the paper introduced the logarithm of per capita GDP, average temperature, precipitation, humidity, sunshine duration, and controlled the time and space fixed effects. (1) (2) and (3) (4) respectively correspond to the parameter estimation before and after grouping. The estimation results showed that the correlation effect coefficients of both the AIR quality index and its index peak did not pass the 10% significance test, indicating that the grouping was reasonable and there was no systematic difference.

In addition, We want to verify that if the policy effects of the top six pollutants in the grouping pass the significance test, it means that if the natural resources audit is not conducted, the six pollutants will show heterogeneity before and after the grouping, which can negate the role of natural resources audit. This paper introduces six major air pollutants for the grouping trend test, and the specific estimated results are shown in Table 3. After parameter estimation, the six major pollutants failed to pass the significance test, indicating that there was no obvious emission reduction effect before the pilot, further indicating the rationality of grouping.

**4.2.2 Placebo test.** The core idea of the placebo test is to make up the processing group or make up the policy time for estimation. If the regression results of the estimators under different fabrication methods are still significant, it means that the original estimation results are likely to be biased, and the changes in our explained variables are likely to be affected by other policy changes or random factors.

For the heterogeneity of the treatment group, this paper selected the mean and peak values of the explained variables (air quality index) and removed the cities with leadership promotion in the areas where the outgoing audit was implemented, and regrouped them for a placebo test. The estimated results are shown in Table 4. It can be seen that although the peak value of air quality once shows significant, after the introduction of control variables, the policy effect

**Table 2. Group inspection of cities before the pilot audit.**

| The variable name | (1) Aqi | (2) Aqi | (3) Mqi | (4) Mqi |
|---|---|---|---|---|
| The policy effect | -0.0529 (-1.0961) | -0.0551 (-0.7327) | -4.2845 (-0.4020) | 3.3527 (0.0960) |
| Control variables | control | control | control | control |
| Time fixed effect | control | control | control | control |
| Urban fixed effect | control | control | control | control |
| Constant term | 4.5691*** (39.4705) | 6.1718* (1.9879) | 168.0509*** (6.4968) | 105.5567 (0.0863) |

Note: ***, ** and * are significance level tests with significance of 1%, 5% and 10% respectively. The brackets are standard errors of the estimated values. Limited by the length of the table, please refer to the appendix and annex for detailed variable parameter estimates.

**Table 3. Grouping trend test of six major air pollutants.**

| The variable name | (1) PM2.5 | (2) PM10 | (3) SO2 | (4) NO2 | (5) CO | (6) O3 |
|---|---|---|---|---|---|---|
| The policy effect | 0.1480 (0.0347) | -3.6198 (-0.5537) | 3.7026 (1.0742) | -1.0047 (-0.3651) | 0.1255 (0.7126) | -4.3921 (-0.6074) |
| Control variables | control | control | control | control | control | control |
| Time fixed effect | control | control | control | control | control | control |
| Urban fixed effect | control | control | control | control | control | control |
| Constant term | 80.0751 (0.5000) | 109.7527 (0.4849) | -360*** (-2.8381) | 32.8158 (0.4354) | 4.9592 (0.6190) | -120 (-0.5238) |

Note: ***, ** and * are significance level tests with significance of 1%, 5% and 10% respectively. The brackets are standard errors of the estimated values. Limited by the length of the table, please refer to the appendix and annex for detailed variable parameter estimates.

is not significant either in the mean value or the peak value of air quality index. At the same time, because the DID model used in this paper includes the lag time, the selection of multi-stage time points during the pilot period is included in the lagging trend. The above estimates show that no similar policy impact is found, which further verifies the regression results of this paper.

**4.2.3 Propensity score matching—Differences-in-Differences (PSW-DID).** PSM-DID model is a combination of propensity score matching model and differences in differences model. PSM is responsible for screening control individuals, and DID is responsible for identifying the impact of policy shocks. PSM can control the differences between groups that are unobservable but do not change over time. For example, the treatment group and the control group come from two failed areas, or the treatment group or the control group use two sets of questionnaires. Different from PSM (PSM assumes that the influencing factors for individuals to choose to participate or not to participate in the project are observable), PSM-DID allows individuals to choose to participate or not to participate in the project to be affected by some unpredictable factors.

PSM-DID is an improvement based on DID. In DID, we believe that there is no individual difference between the two groups, that is, the time effect of the control group is the time effect of the treatment group. In this way, we exclude the total effect of the experimental group from the time effect of the control group, which is the policy effect we studied. PSM-DID takes into account the individual differences between the experimental group and the control group, removes the total effect of the experimental group from the time effect of the control group, and then removes the individual differences between the two groups, so we can get our policy effect. Compared with DID, PSM-DID takes into account the individual differences between the experimental group and the control group; Compared with PSM, PSM-DID can control not only individual differences caused by measurable variables but also individual differences

**Table 4. Placebo test results.**

| The variable name | (1) Aqi | (2) Aqi | (3) Mqi | (4) Mqi |
|---|---|---|---|---|
| The current effect | -6.5046 (-1.0738) | -0.2283 (-0.0286) | -34.9693* (-1.8868) | -1.8798 (-0.0987) |
| Delayed stage effect | 2.1786 (0.3390) | -0.8542 (-0.1050) | 17.5310 (1.4050) | -2.7880 (-0.1771) |
| Control variables | | control | | control |
| Time fixed effect | control | control | control | control |
| Urban fixed effect | control | control | control | control |
| Constant term | 91.9646*** (125.9169) | $1.5 \times 10^3$ (0.9370) | 141.0751*** (51.0046) | $8.9 \times 10^3$ (1.4870) |

Note: ***, ** and * are significance level tests with significance of 1%, 5% and 10% respectively. The brackets are standard errors of the estimated values. Limited by the length of the table, please refer to the appendix and annex for detailed variable parameter estimates.

**Table 5. Estimation of THE PSW-DID model.**

| The variable name | (1) Aqi | (2) Mqi | (3) Citysewage | (4) Citypatent |
|---|---|---|---|---|
| The current effect | 0.0605 (1.0481) | -32.9898* (-1.9290) | -0.0326 (-1.0169) | 0.0687 (1.4877) |
| Delayed stage effect | -0.2290* (-1.9738) | -59.0879*** (-4.2056) | 0.0027 (0.0424) | -0.0761 (-0.7617) |
| Control variables | control | control | control | control |
| Time fixed effect | control | control | control | control |
| Urban fixed effect | control | control | control | control |
| Constant term | -110 (-1.2244) | 7000 (0.5503) | -43.0178 (-1.2461) | 76.8525 (0.6410) |

Note: ***, ** and * are significance level tests with significance of 1%, 5% and 10% respectively. The brackets are standard errors of the estimated values. Limited by the length of the table, please refer to the appendix and annex for detailed variable parameter estimates.

caused by unpredictable factors, such as consumer preferences. In reality, the policy is essentially a non-randomized experiment (or quasi-natural experiment), so the DID method used for policy effect evaluation inevitably has a self-selection bias, while the PSM method can match each treatment group sample to a specific control group sample, making the quasi-natural experiment nearly random. In addition, the "trend assumption" of PSM-DID is easier to meet.

To control the systematic difference between the control group and the treatment group when DID was used, especially the influence grouping of unobservable systemic difference, weighted propensity score matching- differences-in-differences (PSW-DID) was selected for regrouping estimation. This paper computes both the environmental and economic effects of our audit and computes the ATT in that we assign the weights $\hat{y}/(1 + \hat{y})$ to each group and compute the specific scores that match each group. The relevant regression results are shown in Table 5, in which, both the AIR quality index and its peak value have a significant first-stage lag term, and the peak value is still significant in the current period. However, the audit pilot has not significantly improved the sewage treatment capacity and patent application number.

To sum up, when local governments deal with environmental problems, there is a strategic motivation for pollution reduction, rather than substantive pollution control and efficiency improvement. Compared with innovative development performance, leading cadres attach more importance to environmental performance linked with promotion. At the same time, compared with the long-term and complex treatment of water pollution, it pays more attention to the prevention and control of air pollution. However, in the long run, setting up the supervision and assessment mechanism of environmental audit is beneficial to the improvement of environmental quality. In addition, different pollutant attributes lead to different treatment methods and difficulties.

## 5. Further study

### 5.1 Discussion on the mechanism of innovation elements

Both neoclassical economic growth theory and endogenous economic growth theory hold that technological innovation is the key to leading rapid economic growth. No matter Cobb-Douglas production function introduces labor input and material capital input as variables, or Romer model, Lucas model, and Grossman—Chapman model takes manpower and capital as new knowledge resources, the economic growth is inseparable from the input of manpower and material capital. Innovation is essentially the efficient integration of human and capital accumulation, Lucas' new economic growth theory projects technological progress and knowledge accumulation on human capital. Given this, it is necessary to discuss innovation

**Table 6. Discussion of the mechanism of economic performance and manpower level.**

| The variable name | (1) Gdp | (2) Lnpgdp | (3) Popdst | (4) Population |
|---|---|---|---|---|
| The current effect | 0.1946*** ($1.7\times10^3$) | 0.1480** (2.3311) | 0.0468 (0.3440) | 0.0052 (0.8733) |
| Delayed stage effect | 0.3410*** ($2.9\times10^3$) | 0.3154** (2.2239) | 0.1888 (0.8774) | 0.0093 (1.0358) |
| Control variables | control | control | control | control |
| Time fixed effect | control | control | control | control |
| Urban fixed effect | control | control | control | control |
| Constant term | 50.5926*** (505.6118) | -65.3584 (-1.3786) | -53.5747 (-0.5403) | 4.7174 (1.3893) |

Note: ***, ** and * are significance level tests with significance of 1%, 5% and 10% respectively. The brackets are standard errors of the estimated values. Limited by the length of the table, please refer to the appendix and annex for detailed variable parameter estimates.

elements, especially human capital, to further explore the green development effect of audit pilots.

For the environmental performance lag issue above considerations, this paper further takes economic performance (GDP change value) and manpower level (population density and population number) as explanatory variables. As shown in Table 6, after the control time and the urban fixed effects, by establishing the impact of natural resource audit events on economic performance, we find that the occurrence of natural resource audit events can significantly promote local economic performance. Similarly, we find that the impact of natural resource audit events on human resources has not significantly improved the level of local human resources. That is to say, the development of natural resources assets audit has not urged local governments to realize the human resources level of ascension, However, economic performance and environmental performance related to promotion assessment are more significant. This shows that the local government still stays on the surface of strategic behavior, not by the deep meaning of development for long-term substantive implementation. That means the government only considers the incentive of short-term promotion but does not pay attention to the long-term innovative development effect.

## 5.2 Promotion assessment incentive test

The strategic and substantive behaviors of local governments mainly focus on whether they are related to the promotion assessment and incentive of officials. Based on this, this paper discusses the motivation factors and the importance of human resources level in leading cadres' promotion assessment. Among them, promotion probability is a binary variable. The logit model is constructed to investigate the relationship between environmental indicators, economic indicators, human resource indicators, and the promotion probability of officials. The model is set as follows:

$$P = \beta_0 + \beta_1 Cen + \beta_2 Z + \delta_i + \sigma_t + \varepsilon_{it}$$

Among them, $P$ is the probability of promotion of an official. In this paper, the samples of provincial and central leadership cadres promoted from mayor to secretary were selected. $Cen$ represents the core explanatory variable related to promotion assessment, including air quality index, sewage treatment capacity, GDP growth rate (Rgdp), and urban population. $Z$ refers to control variables, including environment, economy, and official characteristic variables, such as humidity, precipitation, CPI, age, education background.

The specific estimation results are shown in Table 7. Equation (1)—(4) is the Logit model, and Equation (5)—(7) is the double fixed-effect model. As you can see, air quality and economic level are associated with the leading cadre promotion. This suggests that adding natural

**Table 7. Discussion on leading cadres' promotion assessment and manpower level.**

| The variable name | (1) Promt | (2) Promt | (3) Promt | (4) Promt | (5) Aqi | (6) Citysewage | (7) Rgdp |
|---|---|---|---|---|---|---|---|
| Aqi | -0.0322* (-1.7439) | | | | | | |
| Citysewage | | -0.2057 (-0.7183) | | | | | |
| Rgdp | | | 3600* (1.7991) | | | | |
| Population | | | | -0.6566 (-1.5503) | 0.1535*** (2.8638) | 0.1742* (1.6561) | 0.3593*** (3.6000) |
| Control variables | control | control | control | control | control | control | control |
| Time fixed effect | control | control | control | control | control | control | control |
| Urban fixed effect | control | control | control | control | control | control | control |
| Constant term | 2.9367 (0.5221) | 288.4970 (0.9223) | 1.9933 (0.2474) | 254.1586 (0.8057) | 1.7573 (0.0637) | 41.2278*** (3.6776) | 11.5723 (0.9056) |

Note: ***, ** and * are significance level tests with significance of 1%, 5% and 10% respectively. The brackets are standard errors of the estimated values. Limited by the length of the table, please refer to the appendix and annex for detailed variable parameter estimates.

resource assets audit and assessment of officials linked policy goals can improve the part of the environment, but the sewage treatment capacity and population levels do not materially affect local officials to the promotion. And it also makes the local officials not improve their enthusiasm for these two aspects. Considering the relevant estimates of population-level, Equations (5)—(7) show that human resources can significantly improve both environmental and economic levels. There is a strategic motive for local governments to improve the environment by simply coping with the "green" supervision of natural resource assets audit, rather than paying full attention to substantive "development" efficiency.

## 6. Research conclusions and policy implications

Although an outgoing audit of natural resource assets is an environmental policy, its implementation not only reflects the effective supervision of environmental problems but also shows the self-consistent disposal mode of local government for environmental protection and economic development. In this paper, we discuss the motivation of Chinese local governments to control pollution, use DID model, and construct PSW-DID to estimate and test robustness. The study finds that when dealing with environmental problems, local governments have a strategic motivation to reduce pollution rather than substantive pollution control and efficiency increase. Compared with the performance of green innovation development, leaders pay more attention to the environmental performance linked to promotion; Compared with the long-term and complexity of water pollution control, it pays more attention to the prevention and control of air pollution; In the long run, adding a supervision and assessment mechanism for environmental audit is indeed beneficial to the improvement of environmental quality, but this policy has a lag period of two years (one lag); In addition, human resources are important both in protecting the environment and promoting the economy. However, the long-term neglect of human capital and the timeliness of environmental supervision have led to the local government's failure to achieve substantial pollution control and efficiency increase through green development. The results show that there is a strategic motivation for local governments to reduce pollution, rather than substantive pollution control and efficiency improvement.

First, in view of the contradiction between the temporary tenure of leading cadres and the long-term governance of environmental problems, the timeliness of audit supervision, and the lag of regulatory effect, it is necessary to ban the characteristics of the time point of departure, establish a long-term mechanism for natural resource assets audit, establish accountability system or change the audit mode as the audit during the tenure, so as to effectively achieve the integration of full coverage and the whole period of audit, and realize the effective

responsibility supervision of leading cadres on environmental quality during their specific tenure. Compared with the long-term and complex treatment of water pollution, it pays more attention to the prevention and control of air pollution. In the long run, the addition of environmental audit 11 supervision and assessment mechanism is indeed beneficial to the improvement of environmental quality, but this policy has a lag period, the effective period is two years (one lag); In addition, human resources play an important role both in protecting the environment and promoting the economy. However, the long-term neglect of human capital and the contradiction between the timeliness of environmental supervision result in the local government not realizing the substantive pollution control and efficiency enhancement of green development.

Secondly, In view of the different attributes of the six pollutants, the difficulty and cycle of treatment are quite different. Given the link mechanism between performance appraisal and promotion, it should not only pay attention to the superficial indicators that are easy to be tested. If leading officials only adopt coping strategies and focus only on pollutants with high performance, environmental quality will not be substantially improved. This not only requires that leading cadres should get rid of the simple consideration of promotion performance but also requires that the audit of natural resources assets should adapt to the new situation of full coverage. This means that it is necessary to fully understand the heterogeneity of audit supervision, with special monitoring for special attributes.

Finally, In view of the importance of human capital in the coordinated development of the environment and economy, natural resource asset audit should not only pay attention to some objective descriptive data in specific fields but also pay attention to substantive characteristic data consistent with the policy, such as the number of population inflow in environmental monitoring. In addition, local governments should promote the rational flow of human resources and realize the substantive pollution control and efficiency enhancement of green development in the process of effective integration of human capital.

Moreover, the limitations of the study are that the data structure is not really able to support the pollution data of individual enterprises, and that large-scale clusters of the same type of production have not yet been formed in various regions of China, and that the pollution situation is not limited to the scope of the enterprises as the weather changes, so we have to choose the region—the prefecture-level city—as the starting point for sample selection.

## Supporting information

**S1 Dataset.**
(XLS)

**S1 Appendix. Descriptive statistics.**
(DOCX)

**S2 Appendix.**
(ZIP)

## Acknowledgments

We thank the editors and the anonymous reviewers for providing thoughtful and valuable comments.

## Author Contributions

**Conceptualization:** Meng Wu.

**Data curation:** Meng Wu.

**Formal analysis:** Meng Wu.

**Methodology:** Guofeng Li, Meng Wu, Ruoyuan Sun.

**Writing – original draft:** Guofeng Li, Meng Wu, Ruoyuan Sun.

**Writing – review & editing:** Guofeng Li, Meng Wu, Ruoyuan Sun.

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
