## [Decision Letter · Decision Letter 0]

15 Sep 2022

PONE-D-22-17190Will China’s audit of natural environmental resource promote green sustainable development? Evidence from PSM-DID analysis based on substantial and strategic pollution reductionPLOS ONE

Dear Dr. Wu,

Thank you for submitting your manuscript to PLOS ONE. After careful consideration, we feel that it has merit but does not fully meet PLOS ONE’s publication criteria as it currently stands. Therefore, we invite you to submit a revised version of the manuscript that addresses the points raised during the review process.

We look forward to receiving your revised manuscript.

Kind regards,

Ashfaque Ahmed Chowdhury, PhD, FHEA, FIEB

Academic Editor

PLOS ONE

Journal Requirements:

Reviewers' comments:

Reviewer's Responses to Questions

**Comments to the Author**

1. Is the manuscript technically sound, and do the data support the conclusions?

Reviewer #1: Yes

Reviewer #2: Yes

2. Has the statistical analysis been performed appropriately and rigorously? 

Reviewer #1: Yes

Reviewer #2: Yes

3. Have the authors made all data underlying the findings in their manuscript fully available?

Reviewer #1: Yes

Reviewer #2: Yes

4. Is the manuscript presented in an intelligible fashion and written in standard English?

Reviewer #1: Yes

Reviewer #2: Yes

5. Review Comments to the Author

Reviewer #1: I think it is an interesting work, which deals with a very timely topic. It is generally well written and structured, and its methodology is appropriate. However, there are some things that should be improved before publication.

In order of appearance rather than importance:

1. Abstract. “In this paper, we established a model of multi-period PSW-DID (weighted propensity score matching-double difference) to test the policy effect using panel data from 2010 to 2017”. What policy effect? What do you mean by that? After reading the body of the paper you understand, but by Reading the abstract you have to.

2. In my view, when reading the Introduction to a paper you have to get much more information. Therefore, I would expand this section so that the reader, after reading it, knows more precisely what the paper is going to deal with.

3. Line 162. The meaning of R&D is well-known.

4. Line 180. “A review of the above works of the literature” Do you mean the four papers mentioned above? Is that all there is?

5. I insist, the reader should know a bit more about previous papers to fully understand the strengths of this one.

6. DID stands for difference-in-differences?

7. Some other economic indicators, such as human capital and a variable capturing industry-mix, could have been used.

8. Data. What is missing is some kind of clear table showing basic quantitative and qualitative data about the research sample.

9. Is per capita income variable included in both economic indicators and control variables?

10. “To avoid heteroscedasticity, some data with high fluctuation are treated with a logarithm.” Not precise enough. The aforementioned Table would clarify that.

11. The ability of time lag to deal with endogeneity is limited.

12. I would include full results in an Appendix.

13. Some of the conclusions offered are, let’s say, quite brave.

14. In my view, you should clarify further each robustness tests. Otherwise, a non-expert reader…

15. The above is obvious for subsection 4.2.3.

16. Sorry, but I cannot understand well the reasoning behind section 5.1. GDP change is more a result of innovation than a condition. Innovation related to population number?

17. The tables have been poorly prepared and are opaque. All necessary information must be included in the own Table, not only in the text.

Reviewer #2: 1. The abstract is very short, and the authors didn't write a clear summary of the goals, key methods, important findings, and conclusions.

2. The authors failed to critically discuss the limitations of existing knowledge in the introduction section.

3. In the introduction, the authors must explain the research gap with the help of prior literature.

4. Research questions are missing.

5. What is the study's significance for China?

6. The content of "Results and discussion" is deficient and lacks a proper explanation of results. The results should be compared and contrasted with prior findings for support and further insight and analysis. Also, the results should be explained from the perspective of the selected economy.

7. All data in this paper is displayed in Table form. Is it possible to convert some tables into graphics so that it becomes easily understood? In some cases, the bar arrangement is easily understood than a list of numbers.

8. Policy implications are insufficient and unimpressive. Please suggest some concrete and relevant policy implications based on the positive and negative shocks of the variables.

9. Please identify and critical elaborate the limitations of the study.

6. PLOS authors have the option to publish the peer review history of their article (what does this mean?). If published, this will include your full peer review and any attached files.

Reviewer #1: No

Reviewer #2: **Yes: **Manzoor Ahmad

---

## [Author Response · Author response to Decision Letter 0]

4 Nov 2022

Dear reviewers and editors, 

First of all, thank you very much for taking the time out of your busy schedule to read and modify my manuscript. Thank you for your valuable suggestions. You have corrected all aspects of the structure, content, research methods and results of my manuscript, which will play a very important role in improving the quality of my manuscript. We have studied comments carefully and have made correction which we hope meet with approval. Revised portion are marked in red in the paper. The main corrections in the paper and the responds to the reviewer’s comments are as flowing: 

Reviewer #1:

1.Comment: 

Abstract. “In this paper, we established a model of multi-period PSW-DID (weighted propensity score matching-double difference) to test the policy effect using panel data from 2010 to 2017”. What policy effect? What do you mean by that? After reading the body of the paper you understand, but by Reading the abstract you have to.

Response and revise: 

Many thanks to the reviewer for the comment of our abstract.It was indeed an oversight on our part to remove the audit event description when adjusting the abstract content, which did cause confusion to the reviewers as well as the readers.

Here, we strongly agree with the reviewers' comment，and we reorganize the abstract content strictly according to the principles of “After reading the body of the paper you understand, but by Reading the abstract you have to”, so that the summary can be detailed and refined.

We revise and add “Countries around the world have become concerned about their responsibility to protect the environment and resources. In this paper, we established a model of multi-period PSW-DID (weighted propensity score matching-double difference) to test the effect of China’s audit of natural resource using panel data from 2010 to 2017” to our paper.Revised portion are marked in red in the paper. 

2.Comment: 

In my view, when reading the Introduction to a paper you have to get much more information. Therefore, I would expand this section so that the reader, after reading it, knows more precisely what the paper is going to deal with.

Response and revise:

The reviewer's comments are very pertinent, and the text does tend to confuse the reader in the introduction, which provides less information about the full text. After discussion, we have reorganized the introduction according to the reviewers' comments and added a description of the full text so that the reviewers and readers can read it and know exactly what the paper will deal with. Due to space limitation, please see the red letter in the script for specific modifications.

3.Comment: 

Line 162. The meaning of R&D is well-known.

Response and revise:

In accordance with the reviewer's comments, we removes excessive interpretation of the significance of R&D in the paper, such as “research and development”.

4. Comment:

Line 180. “A review of the above works of the literature” Do you mean the four papers mentioned above? Is that all there is?

Response and revise:

This opinion put forward by the reviewer made us reflect. It is really because of the problems we expressed in typesetting, which is easy to make readers confused. What we mean by "the above literature review" does not only refer to the above four papers, but actually an overview of the previous papers. That is the whole second chapter "2. Relevant theoretical analysis".

Through the feedback of the reviewer, we adjusted the structure of the chapter, increased the classification of the title “2.5 Above literature analysis”,and reorganize the description to separated the comments into sections to avoid readers' confusion.

5.Comment:

I insist, the reader should know a bit more about previous papers to fully understand the strengths of this one.

Response and revise:

Thanks to the reviewer, we read the literature review again and found that it is really weak. After discussion, we reorganize the literature and add literature content about relevant parts. Due to space limitation, please see the red letter in the script for specific modifications.

6. Comment:

DID stands for difference-in-differences?

Response and revise:

The double difference method is also called "multiplicative difference method", which is abbreviated as DID, from the full English term “Differences-in-Differences”, and has been widely used in policy evaluation in recent years for the following reasons：(1)The endogeneity problem can be avoided to a large extent: policies are generally exogenous with respect to microeconomic agents, and thus there is no reverse causality problem. Moreover, the use of fixed effects estimation alleviates the problem of omitted variable bias to a certain extent. (2) In contrast to the traditional method, which assesses policy effects by setting a dummy variable for the occurrence or non-occurrence of policies and then running a regression, the model setting of the double difference method is more scientific and can estimate the policy effects more accurately. 

Based on the comments of the reviewer, we reinquire the interpretation of DID, add relevant descriptive information in the text, and revised the full name of DID.

7. Comment:

Some other economic indicators, such as human capital and a variable capturing industry-mix, could have been used.

Response and revise:

We put the human capital factor in "5.1 Discussion on the mechanism of innovation elements", because in the discussion of innovation elements, innovation is essentially the efficient integration of human and capital accumulation. Lucas's new economic growth theory focuses on human capital for technological progress and knowledge accumulation. Therefore, we discuss the innovative elements, especially the human capital, to further explore the green development effect of the audit pilot.

In addition, although the green innovation data used in this paper is the total amount of each enterprise, the foothold of the research is in the prefecture level regions, and the natural resources departure audit proposed in this paper is for local government departments. Besides, as far as air pollution is concerned, it is impossible to explore the role of each enterprise in air pollution. Water pollution is only serious in specific enterprises, and selecting industry structure is extremely likely to lead to sample selection deviation.

In fact, the reviewer's suggestion is very good, however, the data structure is really unable to support the pollution data of individual enterprises, and the regions of China have not yet formed large scale clusters of the same type of production, and with the change of weather, the pollution is not limited to the range to which the enterprises belong, so we choose the region - the prefecture-level city - as the sample selection starting point. After repeated discussions, we do not want to waste the reviewer's suggestion, and we place it in "the limitations of the study" to facilitate the follow-up study.Please see the red letter in the script for specific modifications.

8. Comment:

Data. What is missing is some kind of clear table showing basic quantitative and qualitative data about the research sample.

Response and revise:

In response to the comment of the reviewer, we downloaded and read some articles of PLOS ONE again, and found that the information provided in the tables in our articles should be improved. Therefore, we reorganize the tables, retain the compressed form in the text, and put the detailed table information in "Appendix A" at the end of the article, which is the table showing basic quantitative and qualitative data about the research sample.

9.Comment: 

Is per capita income variable included in both economic indicators and control variables?

Response and revise:

We thank the reviewer for seriousness and rigor. Indeed, we use lnpgdp as an economic indicator in the "5. Further study" chapter, where we want to discuss the economic characteristics of the separation, while we introduce lnpgdp in the parameter estimation in the previous chapter to control for economic differences.The research content of "5. Further study" is further deepened, so there are differences in the explanatory and control variables.

In view of the confusion of reviewers and readers, the description of this part is reorganized and modified, redundant variables are deleted, and the correlation is determined. Moreover, the promotion probability of officials in the follow-up study - "5. Further study" (detailed description in "5.2 Promotion assessment incremental test") is added. For details, see the red letter in the paper.

10.Comment: 

 “To avoid heteroscedasticity, some data with high fluctuation are treated with a logarithm.” Not precise enough. The aforementioned Table would clarify that.

Response and revise:

We delete this sentence according to the comments of the reviewer. When explaining the choice of variables in the variable description, the variables in logarithmic form are directly marked in the text, such as "This paper uses the logarithmic form of GDP per capita to describe this indicator".For details, see the red letter in the paper.

11.Comment: 

The ability of time lag to deal with endogeneity is limited.

Response and revise:

Thank the reviewer for the suggestions. The reviewer has rich professional knowledge and insight into endogeneity. It may be that there are many delayed variables used in the article, which is easy to confuse the readers. But the time lag variables in this paper are not mainly used to deal with endogeneity. The time lag variables in this paper are used to explain the time lag of natural resources leaving audit effect.(“ Under normal circumstances, leading cadres will serve for at least two years the first year is the probation period”).

Theoretically, DID can solve the endogenous problem. The principle of DID is to select two groups of similar samples, one is affected by the policy and the other is not. Identify the impact of policies by comparing the differences between the two before and after being affected by policies. Of course, if we require the two groups of samples to be exactly the same, it is too difficult. Even if they are similar, it is also a harsh condition in practical applications. So DID relaxed the requirement that the two groups of samples need not be very similar, as long as they have the same change trend. This is the most important parallel trend assumption of DID. This assumption takes into account both the recognition effectiveness of DID and the practicality of its application. 

Because it is difficult for us to find similar samples in reality, for example, it may be difficult to find enterprises similar to another enterprise and similar individuals, but it is easy to find enterprises or individuals with similar trends. Therefore, we conducted a common trend test (“4.2.1 Trend test”). As long as there is the same change trend, we can reasonably guess that if other conditions remain unchanged, everything will remain unchanged ( the same change trend ). Therefore, the endogenous treatment of DID is that, first of all, we have found two groups of individuals with the same trend. The only difference between the two groups of individuals is that one group is affected by the policy and the other is not. So this is actually close to a randomized trial.Secondly, we also controlled the time and individual fixed effects, so we also excluded those fixed effects. So far, theoretically speaking, there is no endogenous problem. Then, the later PSM selects the processing group and matching group, that is, a D in the DID, and the double difference model needs to do another difference. DID solves the endogenous problem by difference method under the assumption that the conditions are met (the change trend of the experimental group and the control group is the same). In addition, various robustness tests in the following text are also to prove the rationality of DID and overcome endogeneity.

In view of the suggestion of the reviewer, we supplement the above explanations in the paper to clarify the readers' ideas. For details, see the red letter in the paper.

12.Comment: 

I would include full results in an Appendix.

Response and revise:

Thank the reviewer for the suggestion. We provide and update the complete results in the appendix and annexes.

13.Comment: 

Some of the conclusions offered are, let’s say, quite brave.

Response and revise:

As the reviewer says, it is true that some conclusions are brave, but when compared with the reality in China, we can find that the situation seems to be consistent.

At present, there is pressure from local governments for performance assessment, among which the natural environmental resource audit (a kind of environmental protection supervision) does put great pressure on the leaders of local party and government organs.Local governments often have different motives when responding to natural environmental resource audit, and they carry out pollution prevention and control on the surface, but actually achieve little results and do some so-called image projects, which is a strategic purpose to reduce pollution.In addition to responding to pollution prevention audit, local governments also have environmental protection activities for the purpose of obtaining other benefits, which is a strategic behavior, and leadership cadres focus more on environmental protection performance linked to promotion than on green innovation development performance.What can really realize the concept of sustainable development and green development is the actual input of green innovation, which is the so-called substantial efficiency.

This is why we have been calling for and promoting green development, but the effect of pollution control is often poor.

14.Comment:

In my view, you should clarify further each robustness tests. Otherwise, a non-expert reader…

Response and revise:

The reviewer's comments are very pertinent. We give a simple explanation of the assumptions, conditions and problems to be solved for each robustness test in this paper, so that readers can quickly understand the purpose of using the robustness test. For details, see the red letter in the paper.

15.Comment:

The above is obvious for subsection 4.2.3.

Response and revise:

Similar to recommendation 14, We give a simple explanation of the assumptions, conditions and problems to be solved for each robustness test in this paper, so that readers can quickly understand the purpose of using the robustness test. For details, see the red letter in the paper.

16.Comment:

Sorry, but I cannot understand well the reasoning behind section 5.1. GDP change is more a result of innovation than a condition. Innovation related to population number?

Response and revise:

Thank reviewer for the comments. In chapter 5.1, we hope to further explore what factors restrict green innovation and local governments' strategic behavior on the basis of the above research. GDP is the main assessment indicator of local economic performance. We found that the occurrence of natural resource audit events can significantly promote local economic performance by establishing the impact of natural resource audit events on economic performance. In addition, innovation is essentially the efficient integration of human and capital accumulation. We found that the natural resources audit do not significantly improve the level of local human resources. That is to say, in response to environmental supervision, local governments do not pay attention to the role of human resources in green innovation. The implementation of natural resource asset audit do not urge local governments to improve their human resources, but the economic performance linked to promotion assessment was relatively significant.

In view of the suggestion of the reviewer, we further refine this part to avoid similar confusion. Please see the red mark in the paper for details.

17.Comment:

The tables have been poorly prepared and are opaque. All necessary information must be included in the own Table, not only in the text.

Response and revise:

Thank the reviewer for the suggestion. We have listed the complete information of all tables in the text in the appendix and attachment. For details, see the red letter in the paper.

Reviewer #2:

1.Comment: 

The abstract is very short, and the authors didn't write a clear summary of the goals, key methods, important findings, and conclusions.

Response and revise: 

First of all, thanks to the reviewer for the suggestions. We reorganized the summary and integrated the goals, key methods, important findings, and conclusions into the summary. 

The goals: This study can reveal the dilemma of pollution prevention and control in China, urge local governments to promote the rational flow of human resources, base on the improvement of innovation level, and achieve substantive pollution control and efficiency enhancement of green development.

Key methods: PSM-DID.

Findings: (1) local governments had strategic incentive to reduce pollution, they paid more attention to environmental performance linked to individual promotion than to green innovation and development. (2) Compared with the long-term and complexity of water pollution control, they paid more attention to air pollution treatment. (3) In the long run, the environmental audit was indeed beneficial to the improvement of environmental quality, but the time of taking effect was the second year (one-year lag). (4) In addition, because of the contradiction between the neglect of human capital and the timeliness of environmental supervision, the local government did not show substantial pollution reduction.

Conclusions:local officials should foster the long-term responsibility consciousness of green innovation and pay more attention to the integration of human capital. The audit of natural resources should establish a long-term mechanism, which could establish a complete accountability system or change Off-Office audit to Interim audit. The construction of audit big data platform should pay more attention to substantive characteristic data, such as data on population inflow, which is not only a paper score of air pollution. 

For details, see the red letter in the paper.

2.Comment: 

The authors failed to critically discuss the limitations of existing knowledge in the introduction section.

Response and revise:

The reviewer's comments are very pertinent. We add the discussion of the limitations of existing knowledge in the introduction section.For example,”Most researchers have explored the issue from the perspective of pollution control and classification of pollution factors, and few have analyzed the behavior of local governments in response to environmental audits from a motivational perspective. In fact, in addition to responding to the pollution control audit, local governments also engage in environmental protection activities for the purpose of obtaining other benefits, which is a strategic behavior. Studying the motives of different environmental protection has important academic value and policy significance for understanding governmental behavior in the audit implementation process, evaluating the effect of policy implementation, and sorting out the mechanism of action”.

3.Comment: 

In the introduction, the authors must explain the research gap with the help of prior literature.

Response and revise:

According to the suggestions of the reviewer, we added this part,“Academics have put forward many important views on innovative development and environmental protection. Some researchers believe that protecting the ecological environment is to protect productivity, and improving the ecological environment is to develop productivity.Such research links productivity with environmental protection, and innovation is the most critical factor to improve productivity. Another researcher, starting from the influencing factors between the two, believes that green technology innovation is conducive to the emission reduction effect of manufacturing industry in the high human capital group, but the emission reduction effect on the low human capital group is not significant, mainly because human capital and green technology innovation have a certain synergistic effect on reducing environmental pollution. As previously noted, most studies focus on the relationship between innovation and environmental protection or the influencing factors of both.

Although China’s audit of natural resource is an environmental policy, its implementation is not only reflected in the effective regulation of environmental issues, but its operation is more likely to present a harmonious development approach of local governments for environmental protection and economic development. However, local governments in China are subject to environmental and economic performance assessment pressures and tend to show different motivations in dealing with pollution problems. They show strategic pollution reduction motivations rather than substantive pollution control and efficiency, and leadership cadres focus more on promotion-linked environmental performance than on innovative development performance. Most researchers have explored the issue from the perspective of pollution control and classification of pollution factors, and few have analyzed the behavior of local governments in response to environmental audits from a motivational perspective”. 

4. Comment:

Research questions are missing.

Response and revise:

According to the suggestions of the reviewer, we added this part,”From the perspective of motivation, is it strategic or substantive behavior that local governments take when responding to the audit of natural resource? Are the policy effects of audit supervision realized in the current period or in the long term?We will devote ourselves to the discussion of this issue in this study. We will devote ourselves to the discussion of this issue in this study.”

5.Comment:

What is the study's significance for China?

Response and revise:

According to the suggestions of the reviewer, we added this part in introduction,”This paper may contribute in the following aspects: First, studying the effect of environmental audits in guiding local governments to promote environmental protection not only enriches the research on the effect of policy implementation, but also allows us to examine the long-term effect of audits based on the perspective of green development;Secondly, examining the effects of macro policy implementation from the perspective of government motives expands the study of economic policy and government behavior, not only providing new empirical evidence, but also further examining the mechanism of its impact, which helps understanding the transmission mechanism between economic policy to local government behavior; Finally, an analysis based on the presence of government regulation and strong intervention can deepen the understanding of local environmental protection and governance effects”. 

6. Comment:

The content of "Results and discussion" is deficient and lacks a proper explanation of results. The results should be compared and contrasted with prior findings for support and further insight and analysis. Also, the results should be explained from the perspective of the selected economy.

Response and revise:

Thank the reviewer for the suggestions. We make adjustments according to the reviewers' suggestions. Further explain the results from the perspective of China's economy, and sort out and compare the results with previous findings. Limited by space, please see the red mark in the paper for details.

7. Comment:

All data in this paper is displayed in Table form. Is it possible to convert some tables into graphics so that it becomes easily understood? In some cases, the bar arrangement is easily understood than a list of numbers.

Response and revise:

Thanks for the suggestions of the reviewer. Indeed, as the reviewer said, it will be more intuitive to convert some tables into graphics. However, after combing again, we found that the contents shown in the table in our paper are all policy impact effects, and there is no parallel relationship between the effects, and there is no comparative relationship between them. Therefore, although the reviewer's suggestions are very pertinent, we have considered pie chart, straight line chart, histogram, etc., which are not suitable for this study.

8. Comment:

Policy implications are insufficient and unimpressive. Please suggest some concrete and relevant policy implications based on the positive and negative shocks of the variables.

Response and revise:

In response to the comment of the reviewer, we propose some specific and relevant policy impacts according to the positive and negative impacts of variables. please see the red mark in the paper for details.

9.Comment: 

Please identify and critical elaborate the limitations of the study.

Response and revise:

According to the suggestions of the reviewer, we added this part,”Moreover, the limitations of the study are that the data structure is not really able to support the pollution data of individual enterprises, and that large scale clusters of the same type of production have not yet been formed in various regions of China, and that the pollution situation is not limited to the scope of the enterprises as the weather changes, so we have to choose the region - the prefecture-level city - as the starting point for sample selection”.

Finally, thank you again for your guidance, and thank you for reviewing and revising my revised paper again. I hope I can complete an excellent paper with your guidance and help. I sincerely hope my paper can be published in PLOS ONE. 

Sincerely 

Salute!

Li Guofeng, Wu Meng, Sun Ruoyuan

---

## [Decision Letter · Decision Letter 1]

29 Nov 2022

Will China’s audit of natural environmental resource promote green sustainable development? Evidence from PSM-DID analysis based on substantial and strategic pollution reduction

PONE-D-22-17190R1

Dear Dr. Wu,

We’re pleased to inform you that your manuscript has been judged scientifically suitable for publication and will be formally accepted for publication once it meets all outstanding technical requirements.

Kind regards,

Ashfaque Ahmed Chowdhury, Ph.D., FHEA, FIEB

Academic Editor

PLOS ONE

Reviewers' comments:

Reviewer's Responses to Questions

**Comments to the Author**

1. If the authors have adequately addressed your comments raised in a previous round of review and you feel that this manuscript is now acceptable for publication, you may indicate that here to bypass the “Comments to the Author” section, enter your conflict of interest statement in the “Confidential to Editor” section, and submit your "Accept" recommendation.

Reviewer #1: All comments have been addressed

2. Is the manuscript technically sound, and do the data support the conclusions?

Reviewer #1: Yes

3. Has the statistical analysis been performed appropriately and rigorously? 

Reviewer #1: Yes

4. Have the authors made all data underlying the findings in their manuscript fully available?

Reviewer #1: Yes

5. Is the manuscript presented in an intelligible fashion and written in standard English?

Reviewer #1: Yes

6. Review Comments to the Author

Reviewer #1: Thanks for your effort when revising the paper. I believe now it is much clearer and free of misunderstandings

7. PLOS authors have the option to publish the peer review history of their article (what does this mean?). If published, this will include your full peer review and any attached files.

Reviewer #1: No

---

## [Editor Report · Acceptance letter]

2 Dec 2022

PONE-D-22-17190R1 

Will China’s audit of natural environmental resource promote green sustainable development? Evidence from PSM-DID analysis based on substantial and strategic pollution reduction 

Dear Dr. Wu:

I'm pleased to inform you that your manuscript has been deemed suitable for publication in PLOS ONE. Congratulations! Your manuscript is now with our production department. 

Kind regards, 

on behalf of

Dr. Ashfaque Ahmed Chowdhury 

Academic Editor

PLOS ONE